# Estimates of Genetic Parameters, Growth Curve, and Environmental Effects for Nellore Cattle in the Pantanal

**DOI:** 10.3390/vetsci11070318

**Published:** 2024-07-16

**Authors:** Karla Mendonça Vaz, Julio Cesar de Souza, André Luiz Julien Ferraz, Mauricio Vargas da Silveira, Rosana Moreira da Silva de Arruda, Carolina Fregonesi de Souza, Paulo Bahiense Ferraz Filho, Carlos Henrique Cavallari Machado, Mariana Pereira Alencar, Urbano Gomes Pinto de Abreu

**Affiliations:** 1Campus de Aquidauana, Universidade Estadual de Mato Grosso do Sul, Aquidauana 79200-000, MS, Brazil; splinter@uems.br (A.L.J.F.); rosana.mdearruda@gmail.com (R.M.d.S.d.A.); 2Campus de Aquidauana (CPAQ), Universidade Federal de Mato Grosso do Sul, Aquidauana 79200-000, MS, Brazil; julio.souza@ufms.br; 3Ani+ Consultoria e Assessoria Pecuária, Aquidauana 79200-000, MS, Brazil; mauriciozootecnia@gmail.com; 4Programa de Pós-Graduação em Ciências Veterinárias, Faculdade de Medicina Veterinária, Federal University of Uberlândia—UFU, Uberlândia 38400-902, MG, Brazil; carolinasouza727@gmail.com; 5Campus de Três Lagoas, Universidade Federal de Mato Grosso do Sul, Três Lagoas 79603-011, MS, Brazil; paulo.ferraz@ufms.br; 6Faculdades Associadas de Uberaba—UNIUBE, Uberaba 38045-000, MG, Brazil; chcavallari@gmail.com; 7Associação Brasileira de Criadores de Zebu—ABCZ, Uberaba 38045-000, MG, Brazil; marianaalencar18@me.com; 8EMBRAPA/Pantanal, Corumbá 79320-900, MS, Brazil; urbano.abreu@embrapa.br

**Keywords:** weight, selection, *Bos indicus*, Legendre orthogonal polynomials

## Abstract

**Simple Summary:**

The study aimed to estimate growth curves and genetic parameters for Nellore cattle using a random regression methodology. The dataset included 6974 calves raised in Pantanal, MS, Brazil, and 53,233 weight records, with at least four weighings per individual. The model considered direct and maternal genetic additive effects and maternal permanent environmental effects at random. Cubic orthogonal Legendre polynomials were used to fit the growth curve, with fixed effects of sex, year of birth, farm, calf birth month, and cow age at calving. The mean weights at 120 and 205 days were 93.43 kg and 180.42 kg, respectively. Animals born in the dry season had higher average weights, leading to higher weights at 646 days. Direct heritabilities ranged from 0.35 to 0.75, while maternal heritabilities were low (0.03 to 0.08). Random regression effectively adjusts growth curves and aids in selecting superior animals for breeding.

**Abstract:**

The objective was to estimate the growth curves and genetic parameters using random regression methodology for Nellore cattle raised in Pantanal, MS, Brazil (6974 calves; n = 53,233 weights), with at least four weighings per individual. The model considered direct and maternal genetic additives and maternal permanent environmental effects at random. Orthogonal Legendre polynomials of cubic order were used to fit the growth curve. Analyses of variance were performed using the GLM procedure. The model used contained the fixed effects of sex, year of birth, farm, and the covariates calf birth month (linear and quadratic) and cow age at calving (linear and quadratic). The adjusted mean weight at 120 days of age was 93.43 ± 19.78 kg, and for 205 days of age, it was 180.42 ± 26.58 kg. Animals born in the dry season had a higher average weight [kg] (219.57 vs. 211.78, 3.7% higher) and, consequently, had higher weights at 646 days of age. Estimates of direct heritabilities (h^2^a) ranged from 0.35 to 0.75 (high magnitudes), and maternal heritabilities (h^2^m) along the trajectory of low magnitudes ranged from 0.03 to 0.08, respectively. The use of random regression to evaluate beef animals allows for adjusting the growth curve and selecting the best animals to be the parents of future generations.

## 1. Introduction

The wetland Brazilian plain has better environmental conditions for beef cattle [1,2], and through weight performance, it is possible to identify in the herds the animals with the best performance in terms of weight gain and finishing speed [3]. In order to evaluate the genetic parameters of the population, the study was to instruct the breeder in the selection work by recording the weight of the cattle at the different standard ages.

Research involving genetic evaluation produces genetic gain and increases weight gain. These make it possible to reduce the number of days required for a specified finish for the animals. Among the various production scenarios in the beef industry, forecasting the number of days required for the animal to reach slaughter is essential to defining production costs [3,4]. Finding the animals with the greatest genetic potential, which will be slaughtered in the shortest amount of time, is the greatest challenge for the production system. It makes a difference in real economic systems.

Random regression models estimate regression lines for each animal in the pedigree, thereby resulting in the ability to calculate estimated breeding values (EBV) for any age or any number of days on feed. This inherent property of the model allows beef producers to calculate days to finish EBV for finish endpoints that fit individual production scenarios [4,5,6,7].

Evaluating beef cattle using modern technologies has shown good results and significant growth in animal productivity. Identifying the best-fitting models is essential to obtaining a selection response. The use of B-spline polynomials in random regression models for genetic evaluations of beef cattle is important [4]. In beef cattle, animal culling may be associated with multiple biological functions and underlying genetic mechanisms. Correctly estimating the variance components and knowing the different mechanisms that make this possible for different regions are conditions that make the difference when discarding. This is reflected in the estimation of variance components used in genetic and genomic evaluations. Thus, identifying the impact of each culling reason enables estimates of variance components and more robust genetic parameters for animal growth traits and may impact the performance of genetic and genomic evaluations. Avoiding incorrect culling is a fundamental condition for the genetic gain of the herd [3,5,6,7,8,9]. Several statistical models have been used to perform genetic evaluations in beef cattle [3,9,10].

Typically, body weight is an important selection criterion for beef cattle breeding programs [3], which is measured several times over the life of the animal. Through weight performance, it is possible to identify in the herds the animals with the best performance in terms of weight gain and finishing speed. In order to evaluate the genetic parameters of the population, the study was to instruct the breeder in the selection work by recording the weight of the cattle at the different standard ages [11].

Random regression models are often more appropriate for the genetic evaluation of body weight because they describe phenotypic and genetic changes over time. Legendre polynomials are the typical choice for most researchers for fitting mean growth trajectories and additive and permanent environmental effects. Despite the advantages observed in random regression analyses, its use with Legendre polynomials can lead to problems when the variances increase at extremes of age interval [5].

Studying direct and maternal effects on birth weight at 600 days of age [11] reported a decrease in direct and maternal heritability at 150 days (0.14), increasing thereafter. The objective of this study was to estimate the growth curves and genetic parameters for Nellore cattle reared in the Pantanal of Mato Grosso do Sul using a random regression model and estimate the effect of the month of birth on the cows’ weights at the age of 120 and 205 days.

## 2. Materials and Methods

The study used the data of 6974 Nellore animals raised in Mato Grosso do Sul state, Brazil, belonging to the National Zootechnical Archive—Zebu Breeds, provided by the Brazilian Association of Zebu Breeders (ABCZ).

The analyses of variance were performed to evaluate the performance of the calves at 120 (W120) and 205 (W205) days of age using the minimum squares methodology general linear model (GLM) procedure of the statistical analysis system (SAS Student). The model used contained the fixed effects of sex, year of birth, farm, and the covariates calf birth month (linear and quadratic effect) and age of dam at calving (linear and quadratic effect) to study the weights (W120 and W205). The model used was Y = Xβ + e, where: Y = weight vector (W120 or W205) adjusted for the age of the calf; β = fixed effects vector; X = incidence matrix of fixed effects; e = vector of residues, with mean zero and variance = 1; and σ_e_^2^ = component of residual variance.

For random regression analysis, the same dataset was used, and the animals studied had a minimum of four weighings (total of weights evaluated, n = 53,233 information, Figure 1). In this case, the model considered random effects, the direct and maternal additive genetic effects, and permanent and environmental effects of the animal and the mother. The effect of contemporary CG group (sex, farm, birth season, and year of birth of the calf) as a fixed effect and the covariable age of dam at calving (linear and quadratic). The residue was modeled considering eight age classes (in days): 2–91, 92–171, 172–241, 242–316, 317–396, 397–471, 472–551, and 552 and above.

The Legendre orthogonal polynomials of cubic order were used to adjust the population weight growth curve for ages from birth to 700 days. The model has a genetic effect of animal (53,223), sire (536), and dam. The fixed effects of class of animal age (n = 8) on weight; a contemporary group (n = 255) (defined by farm, sex, season of birth (wet season: from October to April; dry season: from May to September) and year of calf birth); and the covariate age of dam, linear and quadratic. To improve the quality of the data file, some restrictions were established: only animals with more than four weight measurements; a minimum of ten animals per age; and, within each contemporary group, a minimum of two bulls and eight animals.

Genetic components and estimates of growth curves were estimated by strict maximum likelihood using WOMBAT Software [12].

## 3. Results

All sources of variation, as well as sex, year of birth, farm, month of birth, and age of cow at calving (linear and quadratic) considered in the model exerted a significant effect (*p* < 0.001) for weights W120 and W205.

The effect of age of dam and birth month on the weight at 120 and 205 days of age of Nellore cattle raised in the Pantanal region of Mato Grosso do Sul, Brazil, was studied. (Figure 2).

The number of dams by age (N) and the respective variations of the average weight (W120 and W205) by the age of the dams are shown in Figure 3.

The number of records and weight averages for each age are shown in Figure 4. A high concentration of records between weaning (205 days of age) and 272 days of age was observed. Weight means increased linearly from birth to 646 days of age, ranging from 31.25 to 345.40 kg, respectively. From 272 days of age, there was a decrease in the number of records.

Data from the random regression analysis with a basis in the birth season of the cattle are presented in Figure 4.

Females had a weight superiority compared to males (Figure 5); up to 212 days of age, on average, they were 6.35 kg heavier than males. At the age of 213 days, males presented a mean of 13.19 kg heavier than females up to 646 days of age, indicating sexual dimorphism.

The segmentation of the heritability curve could be explained by the methodology used, which segmented the classes into ages, thus generating different points of departure. The mean values for the ages of 60 (W60), 120 (W120), 205 (W205), and 365 (W365) are reported in Table 1 and Figure 6 in order to explain the averages of the estimated heritability (direct and maternal) and the permanent environment effect of the animals at 410 (W410), 550 (W550), and 646 (W646) days using the random regression model.

Figure 6 shows the variation of estimates of the additive direct (σ^2^_a_) and maternal (σ^2^_d_) heritability and the permanent environment variance of the animal as a proportion of the total phenotypic variance (σ^2^_pe_), obtained by means of random regression of birth at 646 days of age of Nellore cattle raised in the Pantanal of Mato Grosso do Sul.

## 4. Discussion

The mean weight at 120 days of age adjusted by the least squares means the method was 93.43 ± 19.78 kg, with a coefficient of variation (CV) of 21.16% and a coefficient of determination (R^2^) of 20%. For weight at 205 days of age, the mean was 180.42 ± 26.58 kg, with a coefficient of variation (CV) of 14.73% and a coefficient of determination (R^2^) of 22%. A higher number of dams is observed at 36 months of age, reducing after 144 months. This result corroborates [6,9].

The dams at the beginning of reproductive life between 24 and 48 months produced calves with lower a weight at 120 days of age. The 144-month-old dams showed the maximum point in the production of calves for weight at 120 days of age (W120), with a mean weight of 88.5 kg. In the month of July, calves aged 120 and 205 days presented heavier, with averages of 90.0 kg and 185.3 kg, respectively.

The acceptable performance of Nellore animals reared in the Pantanal region was verified. Ref. [9] studying Nellore animals in different Brazilian regions observed a superiority of males in relation to females of 10.0%, 8.10%, and 7.4%, respectively. Bocchi and Albuquerque [13], studying growth at the age of 120 days, which is the preweaning period in beef cattle, reported that this stage is important due to the ease of obtaining data and also expected greater weight at weaning and the survival of the calf. Dams reach physiological maturity after 60 months of life, increasing milk production and, consequently, calf weight. 

The effect of the dam’s age on weight at W205 showed that the 144-month-old dam presented heavier calves (183.6 kg). From 155 months of age onwards, the dams presented a reduction in the weights of their progenies, with an average of 180.5 kg. The results of the study indicate that dams with advanced aged (144 months) produce heavier calves.

For farmers, it is interesting to carry out well-monitored management and discard these old dams (cows). With this performance management, the farmer not only started to produce heavier calves but also increased the herd renewal rate, increasing the genetic gain of the dams. When discarding older cows, replacements will be made with young heifers with greater genetic potential.

The growth curve according to birth season was a significant source of variation. Animals born in the dry season presented a higher mean weight (219.57 kg vs. 211.78 kg), 3.7% in favor of dry season weights, and, consequently, higher weights at 646 days. The amount of milk produced by the cow is fundamental for the development of the calf until weaning. The genotype implies the development of calves. However, it is important that they have good maternal ability, can produce milk, and wean heavier (above average) calves [9,13].

With random regression, it is possible to observe where there is more variation for growth traits, and, at that moment, the selection will be more efficient, thus identifying higher quality animals [9,14,15].

Estimates of direct additive heritability obtained using a random regression model showed a reduction in their magnitudes from birth (0.40) to 30 days (0.22). After all of the magnitude, there was an increase in the estimates, reaching 0.75 at 646 days of age. This implies that direct selection has shown good results in the region.

Estimates for maternal heritability along the trajectory presented low magnitudes, with higher values (0.08) occurring approximately in the age range between W120 and W205 days, gradually decreasing to 646 days of age (0.03).

Estimates of animal permanent environment variance as a proportion of the total phenotypic variance (p^2^) obtained by means of random regression decreased gradually at 124 days (0.07) and 208 days (0.089). After these ages, there was a trend of increasing estimates according to age, reaching the maximum value (0.15) at 646 days of age.

The direct heritability estimated for W205 was of high magnitude (0.60). For the weight at 550 days of age, direct heritability was high (0.67). These results were high those found by [14] to Nellore cattle.

The segmentation of the heritability curve could be explained by the methodology used, which segmented the classes into ages, thus generating different points of departure. The mean values for the ages of 60 (W60), 120 (W120), 205 (W205), and 365 (W365) are reported in Table 1 and Figure 6. In order to explain the averages of estimated heritability (direct and maternal) and the permanent environment effect on the animals at 410 (W410), 550 (W550), and 646 (W646) days, the random regression model was used. Similar behavior was observed by [13,14].

The direct heritability estimated for weight at 205 days of age was of high magnitude (0.60), close to the values described by [9]. The weight at 205 days of age is of great importance for the breeder to determine up to 50% of the animal’s weight at 550 days of age, according to [8]. Thus, the selection considering the high direct heritability of the weight at 205 days is interesting because of its importance in the final performance. For weight at the age of 550 days, direct heritability was high (0.67). This is a trait of economic interest since it is selected in programs for the genetic improvement of cattle.

## 5. Conclusions

The month of birth influenced calf weights at weaning (W120) and at 205 days (W205). Cows that remain in the herd for long periods are not reserved to incorporate new genetic material. This reduces genetic gain over time as there is no insertion of young, improved animals into the herd. The dry season showed better-performing and heavier animals at 646 days. Direct heritability showed results of high magnitude, reducing the possibility of promoting population gains through selection for the evaluated characteristics. The expected use of regression allowed the adjustment of the growth curve of Nellore animals in the Pantanal, Mato Grosso do Sul, configuring itself as an important tool for adjusting fixed effects and estimating genetic parameters with greater precision.

## Figures and Tables

**Figure 1 vetsci-11-00318-f001:**
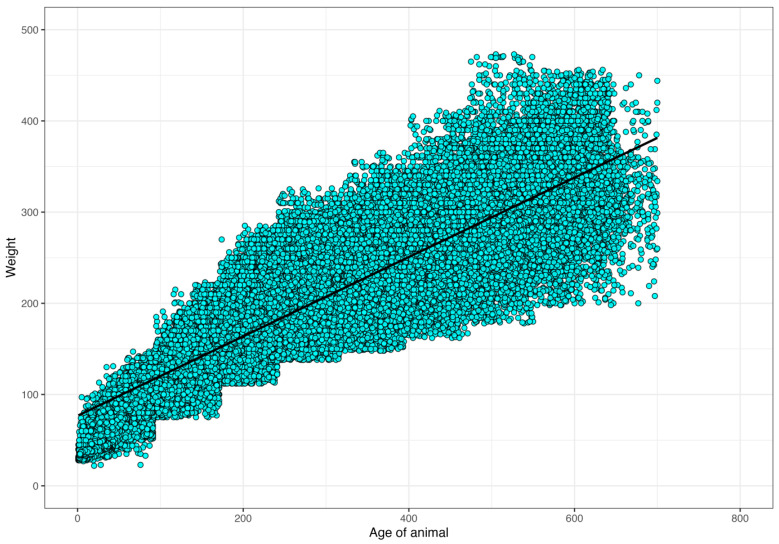
Distribution of weights (n = 53,233) from birth to 700 days of age of Nellore animals reared in the Pantanal region, Brazil (this image is intended to illustrate the data cloud (weights)).

**Figure 2 vetsci-11-00318-f002:**
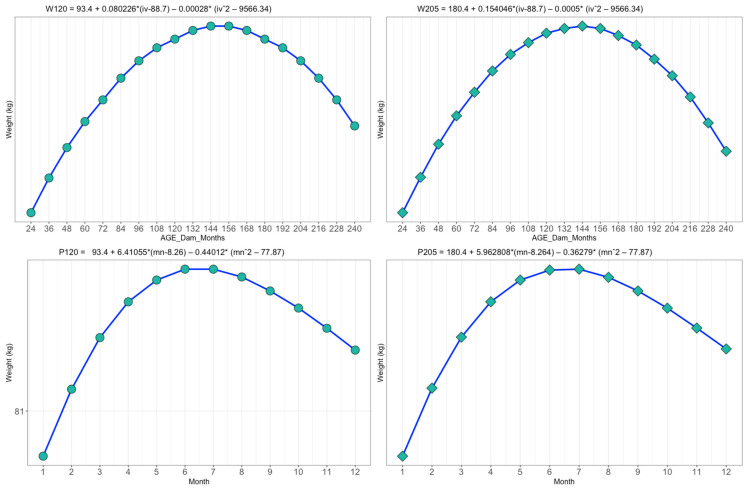
Effect of age of dam and birth month on the weight at 120 and 205 days of age of Nellore cattle raised in the Pantanal region of Mato Grosso do Sul, Brazil.

**Figure 3 vetsci-11-00318-f003:**
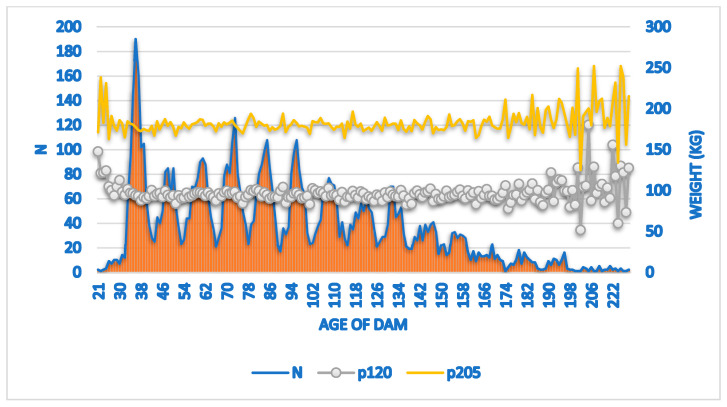
A number of dams (N) and average weight at 120 and 205 days of age of Nellore cattle raised in the Pantanal region of Mato Grosso do Sul.

**Figure 4 vetsci-11-00318-f004:**
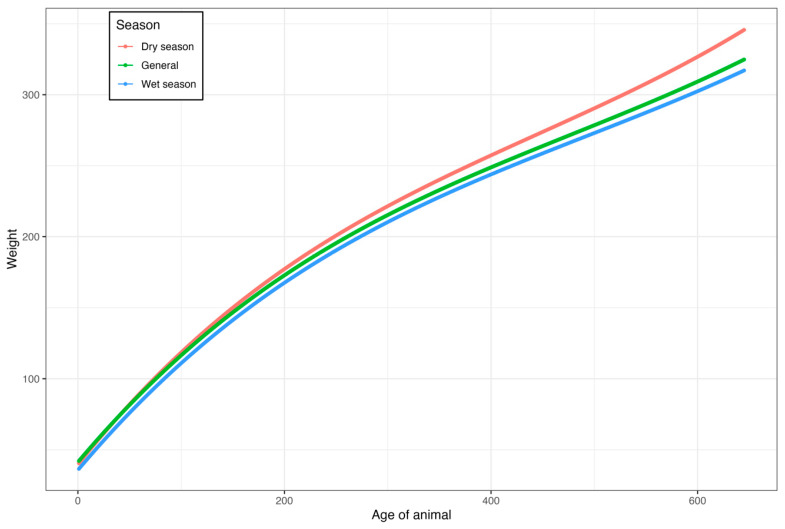
Estimates of the growth curves using a random regression model according to the birth season of cattle raised in the Pantanal of Mato Grosso do Sul.

**Figure 5 vetsci-11-00318-f005:**
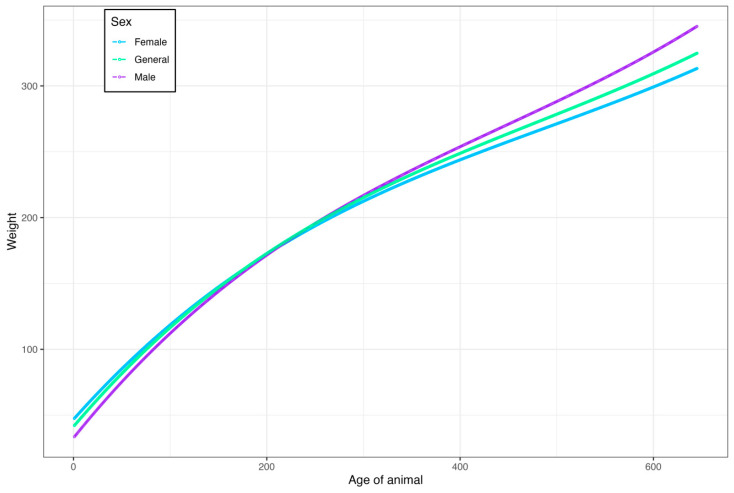
Estimates of growth curves obtained using a random regression model according to the sex of cattle raised in the Pantanal of Mato Grosso do Sul.

**Figure 6 vetsci-11-00318-f006:**
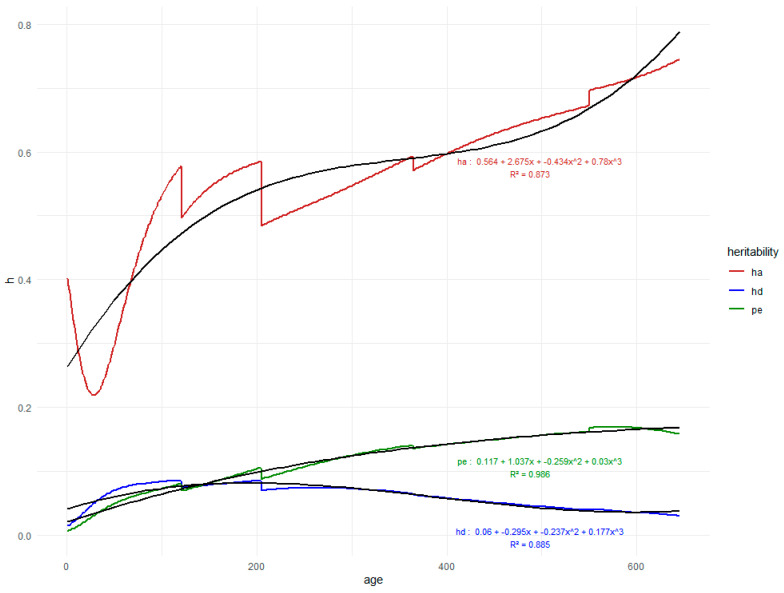
Estimates of heritabilities by age for additive direct (h_a_ = σ^2^_a_) and maternal (h_d_ = σ^2^_m_) heritability and the permanent environment (pe = σ^2^_pe_) of Nellore cattle raised in the Pantanal of Mato Grosso do Sul and polynomial adjusted curve of each heritabilities’ value estimates.

**Table 1 vetsci-11-00318-t001:** Estimation of direct (σ^2^_a_) and maternal (σ^2^_d_) heritabilities, effect of the permanent environment (σ^2^_pe_), and respective standard errors for the weight (in days) of Nellore cattle raised in the Pantanal region, Brazil.

Age	σ^2^_a_	σ^2^_d_	σ^2^_pe_
60	0.35 ± 0.026	0.07 ± 0.015	0.05 ± 0.023
120	0.57 ± 0.020	0.08 ± 0.012	0.08 ± 0.013
205	0.60 ± 0.021	0.08 ± 0.013	0.10 ± 0.012
365	0.60 ± 0.023	0.06 ± 0.012	0.14 ± 0.017
410	0.60 ± 0.023	0.05 ± 0.011	0.14 ± 0.017
550	0.67 ± 0.025	0.04 ± 0.012	0.16 ± 0.021
646	0.75 ± 0.027	0.03 ± 0.014	0.16 ± 0.021

## Data Availability

Data contained within the article.

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
