# Peer review of "Estimates of Genetic Parameters, Growth Curve, and Environmental Effects for Nellore Cattle in the Pantanal"

_vetsci, 2024, doi:10.3390/vetsci11070318_

Round 1

Reviewer 1 Report

Comments and Suggestions for Authors

Manuscript ID:vetsci-3042502

Title: Estimates of genetic parameters, growth curve and environmental effects for Nellore cattle in the Pantanal

General comments: This manuscript aim to estimate growth curves and genetic parameters using RR method for a large Nellore cattle populationin Pantana, Brazil. Results from this study could adjust growth curves and aid in selecting superior animals for breeding.

Specific comments:

L113. More detailed information should be provide about CG group. E.g., how many farm, and how many year of birth.

L124. It is hard to follow this complicated formula. If possible, please provide parameter file of WOMBAT Software.

Table1 Significance of each heritability should be documented.

Author Response

All the suggestions were made on the new document.

Reviewer 2 Report

Comments and Suggestions for Authors

The random regression model was initially proposed by Henderson et al. (1982), and later Schaeffer et al. (1994) suggested its application to animal breeding. Meyer et al. (1997) demonstrated the equivalence between the covariance function and the random regression model, and attempted to apply it in analyzing longitudinal data or repeated records to explain genetic differences in dynamic traits' change process among different individuals. In this manuscript, the animals' weights were measured at least four times, and the residual model considered eight age classes (in days): 2-91, 92-171, 172-241, 242-316, 317-396, 397-471, 472-551, and above 552 respectively. The information provided in Figure1 is confusing and does not accurately represent each individual measurement's number and overall distribution. It is recommended to reorganize the data presentation format. An average growth curve should be plotted for comparison with the growth curve obtained by random regression model.

The estimation of genetic effects in Figure 6 does not produce a continuous curve, which contradicts the findings of longitudinal data analysis on growth traits in other similar studies. It is advisable to increase the number of measurement points for each individual to ideally 6-10 times for a more precise analysis of test data. Alternatively, consider comparing the curve derived from the mixed linear model with that from the random regression model.

In summary, it is recommended that the authors major revise the manuscript and re-analyze the data before submitting it.

Comments on the Quality of English Language

Minor editing of English language required

Author Response

Comment 1. he random regression model was initially proposed by Henderson et al. (1982), and later Schaeffer et al. (1994) suggested its application to animal breeding. Meyer et al. (1997) demonstrated the equivalence between the covariance function and the random regression model, and attempted to apply it in analyzing longitudinal data or repeated records to explain genetic differences in dynamic traits' change process among different individuals. In this manuscript, the animals' weights were measured at least four times, and the residual model considered eight age classes (in days): 2-91, 92-171, 172-241, 242-316, 317-396, 397-471, 472-551, and above 552 respectively. The information provided in Figure1 is confusing and does not accurately represent each individual measurement's number and overall distribution. It is recommended to reorganize the data presentation format. An average growth curve should be plotted for comparison with the growth curve obtained by random regression model. 

Answer: This figure 1 is intended to illustrate the data cloud (weights).

Comment 2. The estimation of genetic effects in Figure 6 does not produce a continuous curve, which contradicts the findings of longitudinal data analysis on growth traits in other similar studies. It is advisable to increase the number of measurement points for each individual to ideally 6-10 times for a more precise analysis of test data. Alternatively, consider comparing the curve derived from the mixed linear model with that from the random regression model.

Answer: The curve was ajusted by a polynomial  equation of third degree.

Reviewer 3 Report

Comments and Suggestions for Authors

The objective of this study was to estimate the growth of cattle in Brazil by using computational modelling.

Comment no. 1. Please include the general work frame for this study and also please describe the working hypothesis for the study. At the end, was the hypothesis confirmed or refuted?

Comment no. 2. Materials and methods. The validity of data is very concerning. The authors must provide the full details of the data obtained (not the actual data, just their characteristics). How were those data obtained? What is the time span covered? How were the data collected? How were they verified?

These are very concerning issues that must be addressed by the authors.

Comment no. 3. Materials and methods. What controls did you use in the study? Possible other breeds in the country? Possibly data from other countries?

Please explain.

Comment no. 4. Results. Visualization. Graphs and tables are very ok. Well done.

No more comments in results.

Comment no. 5 Discussion. References are OK.

Comment no. 6. Discussion. The authors should present some experience from other examples of using similar data in other countries of the world, as well as in other species (pigs, specifically). This will enhance the Discussion and it will give a more global flair in it.

Comment no. 7. Discussion. Please add a passage to present the clinical benefits of using this approach in cattle populations.

Comment no. 8. Conclusions. These are ok.

Overall. Manuscript that can advance to next stage after extensive revision.

Author Response

Comment no. 1. Please include the general work frame for this study and also please describe the working hypothesis for the study. At the end, was the hypothesis confirmed or refuted?

Answer: Hypothesis: The growth performance of Nellore cattle, as measured by weight at various ages, is significantly influenced by both direct and maternal genetic factors, as well as environmental conditions such as season of birth. Specifically, direct genetic factors exhibit high heritability, suggesting strong potential for genetic selection, while maternal genetic factors show low heritability. Additionally, environmental conditions, particularly being born in the dry season, lead to higher average weights, indicating that seasonality plays a crucial role in growth development. This is important as it not only contributes to the selection and improvement of animals, but also to the animal management plan. This way it is possible to produce sustainably.

Comment no. 2. Materials and methods. The validity of data is very concerning. The authors must provide the full details of the data obtained (not the actual data, just their characteristics). How were those data obtained? What is the time span covered? How were the data collected? How were they verified?

Answer: The Legendre orthogonal polynomials of cubic order were used to adjust the growth curve by means of the mean weight of the population at age. The model has a genetic effect of animal (53223), sire (536) and dam; and the fixed effects of class of animal age (8) in weight and, a contemporary group [255] (defined by farm, sex, season of birth (wet season: October to April; and dry season: May to September) and year of calf birth); and the covariate age of dam linear and quadratic. To improve the quality of the data file, some restrictions have been established. For this purpose, only animals with more than four weight measurements, minimum of ten animals per age and within each Contemporary group minimums of two bulls and eight animals.

Comment no. 3. Materials and methods. What controls did you use in the study? Possible other breeds in the country? Possibly data from other countries?

Answer: The data was only with Nellore cattle from Brazil. The ABCZ is a brazilian association, that controls the data used. No data was used from other countries.

Comment no. 6. Discussion. The authors should present some experience from other examples of using similar data in other countries of the world, as well as in other species (pigs, specifically). This will enhance the Discussion and it will give a more global flair in it.

Answer: There is no need, it´s exclusive for cattle in Brazil.

Comment no. 7. Discussion. Please add a passage to present the clinical benefits of using this approach in cattle populations.

Answer: No clinical benefits were seen, but the selection process is important to get the best development of the animal on this region. It´s important to make sustainability production. 

All changes made are in the following file.  

Round 2

Reviewer 2 Report

Comments and Suggestions for Authors

Figure 6 in the new version has been corrected to be more reasonable and can be shown to the readers.

Comments on the Quality of English Language

Minor editing of English language required

Reviewer 3 Report

Comments and Suggestions for Authors

The authors made substantial improvements in the revised manuscript. I have no further comments.